health and disease and epidemiology

corona virus, reported and unreported cases, parameters identification, epidemic mathematical model

**Author for correspondence:**
P. Magal
e-mail: pierre.magal@u-bordeaux.fr

# SI epidemic model applied to COVID-19 data in mainland China

## J. Demongeot[1], Q. Griette[2,3] and P. Magal[2,3]

[1]Department of Medicine, Université Grenoble Alpes, AGEIS EA7407, 38700 La Tronche, France
[2]Department of Medicine, University of Bordeaux, IMB, UMR, 5251, 33400 Talence, France
[3]CNRS, IMB, UMR, 5251, 33400 Talence, France

 PM, 0000-0002-4776-0061

The article is devoted to the parameters identification in the SI model. We consider several methods, starting with an exponential fit to the early cumulative data of SARS-CoV2 in mainland China. The present methodology provides a way to compute the parameters at the early stage of the epidemic. Next, we establish an identifiability result. Then we use the Bernoulli–Verhulst model as a phenomenological model to fit the data and derive some results on the parameters identification. The last part of the paper is devoted to some numerical algorithms to fit a daily piecewise constant rate of transmission.

## 1. Introduction

Estimating the average transmission rate is one of the most crucial challenges in the epidemiology of communicable diseases. This rate conditions the entry into the epidemic phase of the disease and its return to the extinction phase, if it has diminished sufficiently. It is the combination of three factors, one, the coefficient of virulence, linked to the infectious agent (in the case of infectious transmissible diseases), the other, the coefficient of susceptibility, linked to the host (all summarized into the probability of transmission), and also, the number of contacts per unit of time between individuals [1]. The coefficient of virulence may change over time due to mutation over the course of the disease history. The second and third also, if mitigation measures have been taken. This was the case in China from the start of the pandemic [2]. Monitoring the decrease in the average transmission rate is an excellent way to monitor the effectiveness of these mitigation measures. Estimating the rate is therefore a central problem in the fight against epidemics.

The goal of this article is to understand how to compare the SI model to the reported epidemic data and therefore the

model can be used to predict the future evolution of epidemic spread and to test various possible scenarios of social mitigation measures. For $t \geq t_0$, the SI model is the following:

$$\left.\begin{array}{l} S'(t) = -\tau(t)S(t)I(t) \\ I'(t) = \tau(t)S(t)I(t) - \nu I(t), \end{array}\right\} \tag{1.1}$$

and

where $S(t)$ is the number of susceptible and $I(t)$ the number of infectious at time $t$. This system is supplemented by initial data

$$S(t_0) = S_0 \geq 0 \quad \text{and} \quad I(t_0) = I_0 \geq 0. \tag{1.2}$$

In this model, the rate of transmission $\tau(t)$ combines the number of contacts per unit of time and the probability of transmission. The transmission of the pathogen from the infectious to the susceptible individuals is described by a mass action law $\tau(t)\,S(t)\,I(t)$ (which is also the flux of new infectious).

The quantity $1/\nu$ is the average duration of the infectious period and $\nu I(t)$ is the flux of recovering or dying individuals. At the end of the infectious period, we assume that a fraction $f \in (0, 1]$ of the infectious individuals is reported. Let $CR(t)$ be the cumulative number of reported cases. We assume that

$$CR(t) = CR_0 + \nu f\, CI(t), \quad \text{for } t \geq t_0, \tag{1.3}$$

where

$$CI(t) = \int_{t_0}^{t} I(\sigma)\, d\sigma. \tag{1.4}$$

**Assumption 1.1.** *We assume that*

— $S_0 > 0$ *the number of susceptible individuals at time $t_0$ when we start to use the model;*
— $1/\nu > 0$ *the average duration of infectious period;*
— $f > 0$ *the fraction of reported individuals;*

*are known parameters.*

Throughout this paper, the parameter $S_0 = 1.4 \times 10^9$ will be the entire population of mainland China (since COVID-19 is a newly emerging disease). The actual number of susceptibles $S_0$ can be smaller since some individuals can be partially (or totally) immunized by previous infections or other factors. This is also true for SARS-CoV2, even if COVID-19 is a newly emerging disease. In fact, for COVID-19 the level of susceptibility may depend on blood group and genetic lineage. It is indeed suspected that the blood group O is associated with a lower susceptibility to SARS-CoV2 while a gene cluster inherited from Neanderthal has been identified as a risk factor for severe symptoms [3,4].

At the early beginning of the epidemic, the average duration of the infectious period $1/\nu$ is unknown, since the virus has never been investigated in the past. Therefore, at the early beginning of the COVID-19 epidemic, medical doctors and public health scientists used previously estimated average duration of the infectious period to make some public health recommendations. Here we show that the average infectious period is impossible to estimate by using only the time series of reported cases, and must therefore be identified by other means. Actually, with the data of SARS-CoV2 in mainland China, we will fit the cumulative number of the reported case almost perfectly for any non-negative value $1/\nu < 3.3$ days. In the literature, several estimations were obtained: 11 days in [5], 9.5 days in [6], 8 days in [7] and 3.5 days in [8]. The recent survey by Byrne *et al.* [9] focuses on this subject.

---

**Result**

In §3, our analysis shows that:

— It is hopeless to estimate the exact value of the duration of infectiousness by using SI models. Several values of the average duration of the infectious period give the exact same fit to the data.

— We can estimate an upper bound for the duration of infectiousness by using SI models. In the case of SARS-CoV2 in mainland China, this upper bound is 3.3 days.

---

In [10], it is reported that transmission of COVID-19 infection may occur from an infectious individual who is not yet symptomatic. In [11], it is reported that COVID-19-infected individuals generally develop

symptoms, including mild respiratory symptoms and fever, on average 5–6 days after the infection date (with a confidence of 95%, range 1–14 days). In [12], it is reported that the median time prior to symptom onset is 3 days, the shortest 1 day, and the longest 24 days. It is evident that these time periods play an important role in understanding COVID-19 transmission dynamics. Here the fraction of reported individuals $f$ is unknown as well.

---

**Result**

In §3, our analysis shows that:

— It is hopeless to estimate the fraction of reported by using the SI models. Several values for the fraction of reported give the exact same fit to the data.

— We can estimate a lower bound for the fraction of unreported. We obtain $3.83 \times 10^{-5} < f \leq 1$. This lower bound is not significant. Therefore, we can say anything about the fraction of unreported from this class of models.

---

As a consequence, the parameters $1/v$ and $f$ have to be estimated by another method, for instance by a direct survey methodology that should be employed on an appropriated sample in the population in order to evaluate the two parameters.

The goal of this article is to focus on the estimation of the two remaining parameters. Namely, knowing the above-mentioned parameters, we plan to identify

— $I_0$ the initial number of infectious at time $t_0$;
— $\tau(t)$ the rate of transmission at time $t$.

This problem has already been considered in several articles. In the early 1970s, London & Yorke [13,14] already discussed the time-dependent rate of transmission in the context of measles, chickenpox and mumps. More recently, in Wang & Ruan [15] the question of reconstructing the rate of transmission was considered for the 2002–2004 SARS outbreak in China. In Chowell *et al.* [16], a specific form was chosen for the rate of transmission and applied to the Ebola outbreak in Congo. Another approach was also proposed in Smirnova *et al.* [17].

In §2, we will explain how to apply the method introduced in Liu *et al.* [18] to fit the early cumulative data of SARS-CoV2 in China. This method provides a way to compute $I_0$ and $\tau_0 = \tau(t_0)$ at the early stage of the epidemic. In §3, we establish an identifiability result in the spirit of Hadeler [19].

In §4, we use the Bernoulli–Verhulst model as a phenomenological model to describe the data. As it was observed in several articles, the data from mainland China (and other countries as well) can be fitted very well by using this model. As a consequence, we will obtain an explicit formula for $\tau(t)$ and $I_0$ expressed as a function of the parameters of the Bernoulli–Verhulst model and the remaining parameters of the SI model. This approach gives a very good description of this set of data. The disadvantage of this approach is that it requires an evaluation of the final size $CR_\infty$ from the early beginning (or at least it requires an estimation of this quantity).

Therefore, in order to be predictive, we will explore in the remaining sections of the paper the possibility of constructing a day-by-day rate of transmission. Here we should refer to Bakhta *et al.* [20] where another novel forecasting method was proposed.

In §5, we will prove that the daily cumulative data can be approached perfectly by at most one sequence of day-by-day piecewise constant transmission rates. In §6, we propose a numerical method to compute such a (piecewise constant) rate of transmission. Section 7 is devoted to the discussion, and we will present some figures showing the daily basic reproduction number for the COVID-19 outbreak in mainland China.

## 2. Estimating $\tau(t_0)$ and $I_0$ at the early stage of the epidemic

In this section, we apply the method presented in [21] to the SI model. At the early stage of the epidemic, we can assume that $S(t)$ is almost constant and equal to $S_0$. We can also assume that $\tau(t)$ remains constant equal to $\tau_0 = \tau(t_0)$. Therefore, by replacing these parameters into the $I$-equation of system (1.1) we obtain

$$I'(t) = (\tau_0 S_0 - v)I(t).$$

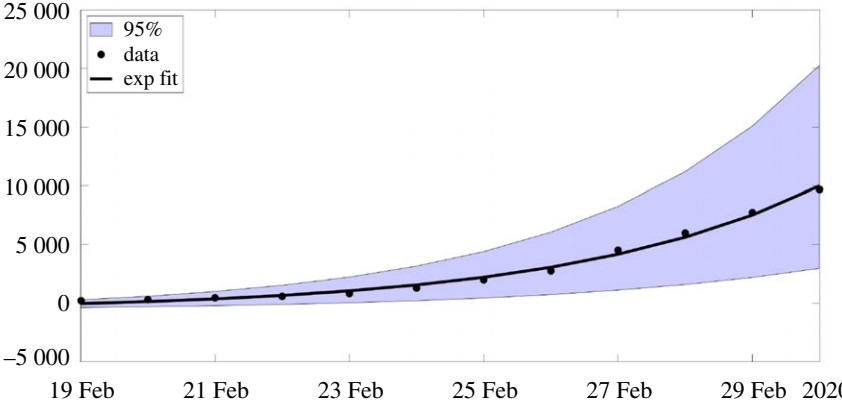

**Figure 1.** In this figure, we plot the best fit of the exponential model to the cumulative number of reported cases of COVID-19 in mainland China between 19 February and 1 March. We obtain $\chi_1 = 3.7366$, $\chi_2 = 0.2650$ and $\chi_3 = 615.41$ with $t_0 = 19$ Feb. The parameter $\chi_3$ is obtained by minimizing the error between the best exponential fit and the data.

Therefore,

$$I(t) = I_0 \exp(\chi_2(t - t_0)),$$

where

$$\chi_2 = \tau_0 S_0 - \nu. \tag{2.1}$$

By using (1.3), we obtain

$$CR(t) = CR_0 + \nu f I_0 \frac{e^{\chi_2(t-t_0)} - 1}{\chi_2}. \tag{2.2}$$

We obtain a first phenomenological model for the cumulative number of reported cases (valid only at the early stage of the epidemic)

$$CR(t) = \chi_1 \, e^{\chi_2 t} - \chi_3. \tag{2.3}$$

In figure 1, we compare the model to the COVID-19 data for mainland China. The data used in the article are taken from [22–24] and reported in appendix A. In order to estimate the parameter $\chi_3$, we minimize the distance between $CR_{Data}(t) + \chi_3$ and the best exponential fit $t \to \chi_1 e^{\chi_2 t}$ (i.e. we use the Matlab function fit(t, data,'exp1')).

---

**The estimated initial number of infected and transmission rate**

By using (1.3) and (2.3), we obtain

$$I_0 = \frac{CR'(t_0)}{\nu f} = \frac{\chi_1 \chi_2 e^{\chi_2 t_0}}{\nu f}, \tag{2.4}$$

and by using (2.1)

$$\tau_0 = \frac{\chi_2 + \nu}{S_0}. \tag{2.5}$$

---

**Remark 2.1.** Fixing $f = 0.5$ and $\nu = 0.2$, we obtain

$$I_0 = 3.7366 \times 0.2650 \times \frac{\exp(0.2650 \times 19)}{(0.2 \times 0.5)} = 1521$$

and

$$\tau_0 = \frac{0.2650 + 0.2}{1.4 \times 10^9} = 3.3214 \times 10^{-10}.$$

The influence of the errors made in the estimations (at the early stage of the epidemic) has been considered in the recent article by Roda *et al.* [25]. To understand this problem, let us first consider the case of the rate of transmission $\tau(t) = \tau_0$ in the model (1.1). In that case (1.1) becomes

and
$$\left.\begin{array}{l} S'(t) = -\tau_0 S(t)I(t) \\ I'(t) = \tau_0 S(t)I(t) - \nu I(t). \end{array}\right\} \qquad (2.6)$$

By using the S-equation of model (2.6) we obtain

$$S(t) = S_0 \exp\left(-\tau_0 \int_{t_0}^{t} I(\sigma)\mathrm{d}\sigma\right) = S_0 \exp\left(-\tau_0 \mathrm{CI}(t)\right),$$

where $\mathrm{CI}(t)$ is the cumulated number of infectious individuals. Substituting $S(t)$ by this formula in the *I*-equation of (2.6) we obtain

$$I'(t) = S_0 \exp\left(-\tau_0 \mathrm{CI}(t)\right)\tau_0 \mathrm{CI}'(t) - \nu I(t).$$

Therefore, by integrating the above equation between $t$ and $t_0$ we obtain

$$\mathrm{CI}'(t) = I_0 + S_0[1 - \exp\left(-\tau_0 \mathrm{CI}(t)\right)] - \nu \mathrm{CI}(t). \qquad (2.7)$$

Remarkably, equation (2.7) is monotone. We refer to Smith [26] for a comprehensive presentation on monotone systems. By applying a comparison principle to (2.7), we are in a position to confirm the intuition about epidemics SI models. Note that the monotone properties are only true for the cumulative number of infectious (this is false for the number of infectious).

**Theorem 2.2.** *Let* $t > t_0$ *be fixed. The cumulative number of infectious* $\mathrm{CI}(t)$ *is strictly increasing with respect to the following quantities*

(i) $I_0 > 0$ *the initial number of infectious individuals;*
(ii) $S_0 > 0$ *the initial number of susceptible individuals;*
(iii) $\tau > 0$ *the transmission rate;*
(iv) $1/\nu > 0$ *the average duration of the infectiousness period.*

---

**Error in the estimated initial number of infected and transmission rate**

Assume that the parameters $\chi_1$ and $\chi_2$ are estimated with a 95% confidence interval

and
$$\chi^-_{1,95\%} \leq \chi_1 \leq \chi^+_{1,95\%}$$

We obtain
$$\chi^-_{2,95\%} \leq \chi_2 \leq \chi^+_{2,95\%}.$$

and
$$I^-_{0,95\%} := \frac{\chi^-_{1,95\%}\,\chi^-_{2,95\%}\,\mathrm{e}^{\chi^-_{2,95\%}\,t_0}}{\nu f} \leq I_0 \leq I^+_{0,95\%} := \frac{\chi^+_{1,95\%}\,\chi^+_{2,95\%}\,\mathrm{e}^{\chi^+_{2,95\%}\,t_0}}{\nu f} \qquad (2.8)$$

$$\tau^-_{0,95\%} := \frac{\chi^-_{2,95\%} + \nu}{S_0} \leq \tau_0 \leq \tau^+_{0,95\%} := \frac{\chi^+_{2,95\%} + \nu}{S_0}. \qquad (2.9)$$

---

**Remark 2.3.** By using the data for mainland China, we obtain

$$\chi^-_{1,95\%} = 1.57,\ \chi^+_{1,95\%} = 5.89,\ \chi^-_{2,95\%} = 0.24,\ \chi^+_{2,95\%} = 0.28. \qquad (2.10)$$

In figure 2, we plot the upper and lower solutions $\mathrm{CR}^+(t)$ (obtained by using $I_0 = I^+_{0,95\%}$ and $\tau_0 = \tau^+_{0,95\%}$) and $\mathrm{CR}^-(t)$ (obtained by using $I_0 = I^-_{0,95\%}$ and $\tau_0 = \tau^-_{0,95\%}$) corresponding to the blue region and the black curve corresponds to the best estimated value $I_0 = 1521$ and $\tau_0 = 3.3214 \times 10^{-10}$.

Recall that the final size of the epidemic corresponds to the positive equilibrium of (2.7)

$$0 = I_0 + S_0[1 - \exp\left(-\tau_0 \mathrm{CI}_\infty\right)] - \nu \mathrm{CI}_\infty. \qquad (2.11)$$

In figure 2, the changes in the parameters $I_0$ and $\tau_0$ (in (2.8) and (2.9)) do not affect significantly the final size.

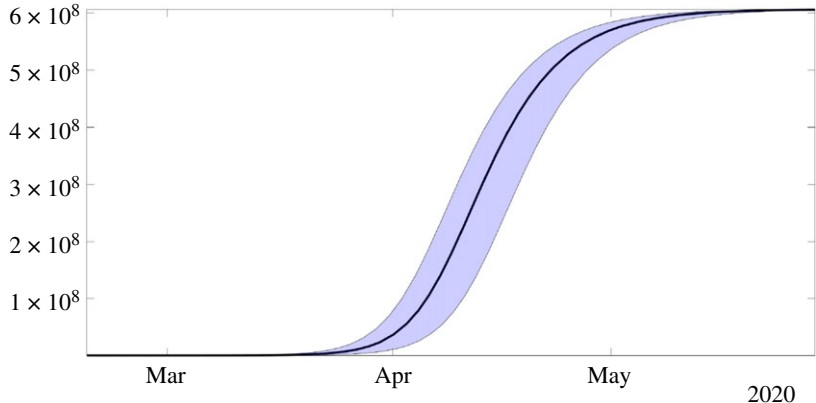

**Figure 2.** In this figure, the black curve corresponds to the cumulative number of reported cases CR(t) obtained from the model (2.6) with CR'(t) = νf I(t) by using the values $I_0 = 1521$ and $\tau_0 = 3.32 \times 10^{-10}$ obtained from our method and the early data from 19 February to 1 March. The blue region corresponds to the 95% confidence interval when the rate of transmission τ(t) is constant and equal to the estimated value $\tau_0 = 3.32 \times 10^{-10}$.

# 3. Theoretical formula for $\tau(t)$

By using the *S*-equation of model (1.1) we obtain

$$S(t) = S_0 \exp\left(-\int_{t_0}^{t} \tau(\sigma)\, I(\sigma)\, d\sigma\right),$$

next by using the I-equation of model (1.1) we obtain

$$I'(t) = S_0 \exp\left(-\int_{t_0}^{t} \tau(\sigma)\, I(\sigma) d\sigma\right) \tau(t)\, I(t) - \nu I(t),$$

and by taking the integral between $t$ and $t_0$ we obtain a Volterra integral equation for the cumulative number of infectious

$$\mathrm{CI}'(t) = I_0 + S_0\left[1 - \exp\left(-\int_{t_0}^{t} \tau(\sigma)\, I(\sigma)\, d\sigma\right)\right] - \nu \mathrm{CI}(t), \tag{3.1}$$

which is equivalent to (by using (1.3))

$$\mathrm{CR}'(t) = \nu f\left(I_0 + S_0\left[1 - \exp\left(-\frac{1}{\nu f}\int_{t_0}^{t} \tau(\sigma)\, \mathrm{CR}'(\sigma) d\sigma\right)\right]\right) + \nu \mathrm{CR}_0 - \nu \mathrm{CR}(t). \tag{3.2}$$

The following result permits to obtain a perfect match between the SI model and the time-dependent rate of transmission τ(t).

**Theorem 3.1.** *Let $S_0$, ν, f, $I_0 > 0$ and $\mathrm{CR}_0 \geq 0$ be given. Let $t \to I(t)$ be the second component of system (1.1). Let $\widehat{\mathrm{CR}}:[t_0, \infty) \to \mathbb{R}$ be a two times continuously differentiable function satisfying*

$$\widehat{\mathrm{CR}}(t_0) = \mathrm{CR}_0, \tag{3.3}$$

$$\widehat{\mathrm{CR}}'(t_0) = \nu f I_0, \tag{3.4}$$

$$\widehat{\mathrm{CR}}'(t) > 0, \forall t \geq t_0 \tag{3.5}$$

and

$$\nu f(I_0 + S_0) - \widehat{\mathrm{CR}}'(t) - \nu(\widehat{\mathrm{CR}}(t) - \mathrm{CR}_0) > 0, \forall t \geq t_0. \tag{3.6}$$

*Then*

$$\widehat{CR}(t) = CR_0 + vf \int_{t_0}^{t} I(s)\mathrm{d}s, \forall t \geq t_0, \tag{3.7}$$

*if and only if*

$$\tau(t) = \frac{vf(\widehat{CR}''(t)/\widehat{CR}'(t) + v)}{vf(I_0 + S_0) - \widehat{CR}'(t) - v(\widehat{CR}(t) - CR_0)}. \tag{3.8}$$

*Proof.* Assume first (3.7) is satisfied. Then by using equation (3.1) we deduce that

$$S_0 \exp\left(-\int_{t_0}^{t} \tau(\sigma)I(\sigma)\mathrm{d}\sigma\right) = I_0 + S_0 - I(t) - vCI(t).$$

Therefore,

$$\int_{t_0}^{t} \tau(\sigma)I(\sigma)\,\mathrm{d}\sigma = \ln\left[\frac{S_0}{I_0 + S_0 - I(t) - vCI(t)}\right] = \ln(S_0) - \ln[I_0 + S_0 - I(t) - vCI(t)]$$

therefore by taking the derivative on both sides

$$\tau(t)I(t) = \frac{I'(t) + vI(t)}{I_0 + S_0 - I(t) - vCI(t)} \Leftrightarrow \tau(t) = \frac{(I'(t)/I(t)) + v}{I_0 + S_0 - I(t) - vCI(t)} \tag{3.9}$$

and by using the fact that $CR(t) - CR_0 = vfCI(t)$ we obtain (3.8).

Conversely, assume that $\tau(t)$ is given by (3.8). Then if we define $\widetilde{I}(t) = \widehat{CR}'(t)/vf$ and $\widetilde{CI}(t) = (\widehat{CR}(t) - CR_0)/vf$, by using (3.3) we deduce that

$$\widetilde{CI}(t) = \int_{t_0}^{t} \widetilde{I}(\sigma)\,\mathrm{d}\sigma,$$

and by using (3.4)

$$\widetilde{I}(t_0) = I_0. \tag{3.10}$$

Moreover from (3.8), we deduce that $\widetilde{I}(t)$ satisfies (3.9). By using (3.10), we deduce that $t \to \widetilde{CI}(t)$ is a solution of (3.1). By uniqueness of the solution of (3.1), we deduce that $\widetilde{CI}(t) = CI(t), \forall t \geq t_0$ or equivalently $CR(t) = CR_0 + vf \int_{t_0}^{t} I(s)\mathrm{d}s, \forall t \geq t_0$. The proof is completed. ∎

Formula (3.8) was already obtained by Hadeler ([19], see corollary 2).

# 4. Explicit formula for $\tau(t)$ and $I_0$

Many phenomenological models have been compared to the data during the first phase of the COVID-19 outbreak. We refer to the paper of Tsoularis & Wallace [27] for a nice survey on the generalized logistic equations. Let us consider here for example, the Bernoulli–Verhulst equation

$$CR'(t) = \chi_2 CR(t)\left(1 - \left(\frac{CR(t)}{CR_\infty}\right)^\theta\right), \forall t \geq t_0, \tag{4.1}$$

supplemented with the initial data

$$CR(t_0) = CR_0 \geq 0.$$

Let us recall the explicit formula for the solution of (4.1)

$$CR(t) = \frac{e^{\chi_2(t-t_0)}CR_0}{\left[1 + (\chi_2\theta/CR_\infty^\theta)\int_{t_0}^{t}(e^{\chi_2(\sigma-t_0)}CR_0)^\theta\,\mathrm{d}\sigma\right]^{1/\theta}} = \frac{e^{\chi_2(t-t_0)}CR_0}{[1 + (CR_0^\theta/CR_\infty^\theta)(e^{\chi_2\theta(t-t_0)} - 1)]^{1/\theta}}. \tag{4.2}$$

**Assumption 4.1.** *We assume that the cumulative numbers of reported cases* $CR_{\mathrm{Data}}(t_i)$ *are known for a sequence of times* $t_0 < t_1 < \cdots < t_{n+1}$ *(see figure 3).*

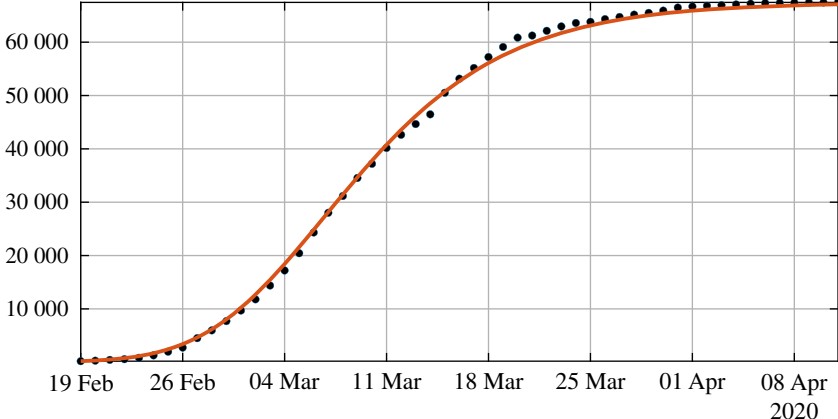

**Figure 3.** In this figure, we plot the best fit of the Bernoulli–Verhulst model to the cumulative number of reported cases of COVID-19 in China. We obtain $\chi_2 = 0.66$ and $\theta = 0.22$. The black dots correspond to data for the cumulative number of reported cases and the red curve corresponds to the model.

---

**Estimated initial number of infected**

By combining (1.3) and the Bernoulli–Verhulst equation (4.1) for $t \to \mathrm{CR}(t)$, we deduce the initial number of infected

$$I_0 = \frac{\mathrm{CR}'(t_0)}{\nu f} = \frac{\chi_2\,\mathrm{CR}_0(1 - (\mathrm{CR}_0/\mathrm{CR}_\infty)^\theta)}{\nu f}. \tag{4.3}$$

---

**Remark 4.2.** We fix $f = 0.5$, from the COVID-19 data in mainland China and formula (4.3) (with $\mathrm{CR}_0 = 198$), we obtain

$$I_0 = 1909 \text{ for } \nu = 0.1$$

and

$$I_0 = 954 \text{ for } \nu = 0.2.$$

By using (4.1), we deduce that

$$\mathrm{CR}''(t) = \chi_2\,\mathrm{CR}'(t)\left(1 - \left(\frac{\mathrm{CR}(t)}{\mathrm{CR}_\infty}\right)^\theta\right) - \frac{\chi_2\,\theta}{\mathrm{CR}_\infty^\theta}\,\mathrm{CR}(t)(\mathrm{CR}(t))^{\theta-1}\mathrm{CR}'(t)$$

$$= \chi_2\,\mathrm{CR}'(t)\left(1 - \left(\frac{\mathrm{CR}(t)}{\mathrm{CR}_\infty}\right)^\theta\right) - \frac{\chi_2\,\theta}{\mathrm{CR}_\infty^\theta}\,(\mathrm{CR}(t))^\theta\mathrm{CR}'(t),$$

therefore

$$\mathrm{CR}''(t) = \chi_2\,\mathrm{CR}'(t)\left(1 - (1+\theta)\left(\frac{\mathrm{CR}(t)}{\mathrm{CR}_\infty}\right)^\theta\right). \tag{4.4}$$

---

**Estimated rate of transmission**

By using the Bernoulli–Verhulst equation (4.1) and substituting (4.4) in (3.8), we obtain

$$\tau(t) = \frac{\nu f(\chi_2\,(1 - (1+\theta)(\mathrm{CR}(t)/\mathrm{CR}_\infty)^\theta) + \nu)}{\nu f(I_0 + S_0) + \nu\mathrm{CR}_0 - \mathrm{CR}(t)(\chi_2(1 - (\mathrm{CR}(t)/\mathrm{CR}_\infty)^\theta) + \nu)}. \tag{4.5}$$

This formula (4.5) combined with (4.2) gives an explicit formula for the rate of transmission.

---

Since $\mathrm{CR}(t) < \mathrm{CR}_\infty$, by considering the sign of the numerator and the denominator of (4.5), we obtain the following proposition.

(a)

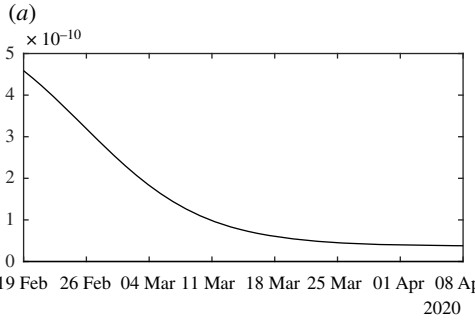

(b)

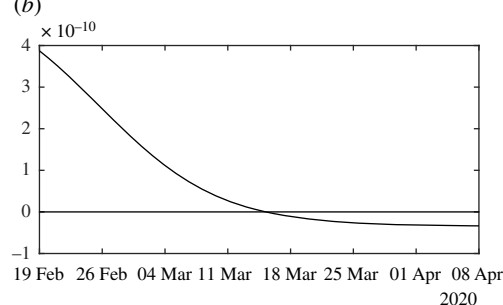

**Figure 4.** In this figure, we plot the rate of transmission obtained from formula (4.5) with $f = 0.5$, $\chi_2\,\theta = 0.145 < \nu = 0.2$ (in (a)) and $\nu = 0.1 < \chi_2\,\theta = 0.145$ (in (b)), $\chi_2 = 0.66$ and $\theta = 0.22$, and $CR_\infty = 67\,102$, which is the latest value obtained from the cumulative number of reported cases for China.

**Proposition 4.3.** *The rate of transmission $\tau(t)$ given by (4.5) is non-negative for all $t \geq t_0$ if*

$$\nu \geq \chi_2\,\theta \tag{4.6}$$

*and*

$$f(I_0 + S_0) + \nu CR_0 > CR_\infty(\chi_2 + \nu). \tag{4.7}$$

---

**Compatibility of the model SI with the COVID-19 data for mainland China**

The model SI is compatible with the data only when $\tau(t)$ stays positive for all $t \geq t_0$. From our estimation of the Chinese's COVID-19 data, we obtain $\chi_2\,\theta = 0.14$. Therefore from (4.6), we deduce that model is compatible with the data only when

$$1/\nu \leq \frac{1}{0.14} = 3.3\,\text{days}. \tag{4.8}$$

This means that the average duration of infectious period $1/\nu$ must be shorter than 3.3 days.

Similarly, the condition (4.7) implies

$$f \geq \frac{CR_\infty\chi_2 + (CR_\infty - CR_0)\nu}{S_0 + I_0} \geq \frac{CR_\infty\chi_2 + (CR_\infty - CR_0)\chi_2\,\theta}{S_0 + I_0}$$

and since we have $CR_0 = 198$ and $CR_\infty = 67\,102$, we obtain

$$f \geq \frac{67\,102 \times 0.66 + (67\,102 - 198) \times 0.14}{1.4 \times 10^9} \geq 3.83 \times 10^{-5}. \tag{4.9}$$

So according to this estimation the fraction of unreported $0 < f \leq 1$ can be almost as small as we want.

---

Figure 4 illustrates proposition 4.3. We observe that the formula for the rate of transmission (4.5) becomes negative whenever $\nu < \chi_2\theta$. In figure 5, we plot the numerical simulation obtained from (1.1) to (1.3) when $t \to \tau(t)$ is replaced by the explicit formula (4.5). It is surprising that we can reproduce perfectly the original Bernoulli–Verhulst even when $\tau(t)$ becomes negative (see figure 3). This was not guaranteed at first, since the I-class of individuals is losing some individuals which are recovering.

# 5. Computing numerically a day-by-day piecewise constant rate of transmission

**Assumption 5.1.** *We assume that the rate of transmission $\tau(t)$ is piecewise constant and for each $i = 0, \ldots, n$,*

$$\tau(t) = \tau_i, \text{ whenever } t_i \leq t < t_{i+1}. \tag{5.1}$$

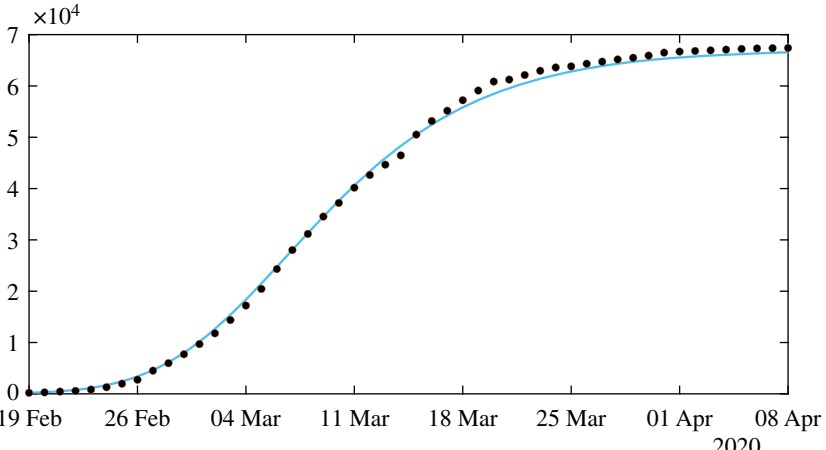

**Figure 5.** In this figure, we plot the number of reported cases by using model (1.1) and (1.3), with the rate of transmission obtained in (4.5). The parameters values are $f = 0.5$, $\nu = 0.1$ or $\nu = 0.2$, $\chi_2 = 0.66$ and $\theta = 0.22$, and $CR_\infty = 67\,102$ is the latest value obtained from the cumulative number of reported cases for China. Furthermore, we use $S_0 = 1.4 \times 10^9$ for the total population of China and $I_0 = 954$ which is obtained from formula (4.3). The black dots correspond to observed data for the cumulative number of reported cases and the blue curve corresponds to the model.

For $t \in [t_{i-1}, t_i]$, we deduce by using assumption 5.1 that

$$\int_{t_0}^{t} \tau(\sigma)\, \mathrm{CR}'(\sigma)\, \mathrm{d}\sigma = \sum_{j=0}^{i-2} \int_{t_j}^{t_{j+1}} \tau_j\, \mathrm{CR}'(\sigma)\, \mathrm{d}\sigma + \int_{t_{i-1}}^{t} \tau_{i-1}\, \mathrm{CR}'(\sigma)\, \mathrm{d}\sigma.$$

Therefore by using (3.2), for $t \in [t_{i-1}, t_i]$, we obtain

$$\mathrm{CR}'(t) = \nu f\left(I_0 + S_0\left[1 - \Pi_{i-1} \exp\left(-\frac{\tau_{i-1}}{\nu f}[\mathrm{CR}(t) - \mathrm{CR}(t_{i-1})]\right)\right]\right) + \nu\, \mathrm{CR}_0 - \nu\mathrm{CR}(t), \tag{5.2}$$

where

$$\Pi_{i-1} = \exp\left(-\sum_{j=0}^{i-2} \frac{\tau_j}{\nu f}[\mathrm{CR}(t_{j+1}) - \mathrm{CR}(t_j)]\right). \tag{5.3}$$

By fixing $\tau_{i-1} = 0$ on the right-hand side of (5.2), we get

$$\mathrm{CR}'(t) \geq \nu f(I_0 + S_0[1 - \Pi_{i-1}]) + \nu\, \mathrm{CR}_0 - \nu\mathrm{CR}(t),$$

and when $\tau_{i-1} \to \infty$ we obtain

$$\mathrm{CR}'(t) \leq \nu f(I_0 + S_0) + \nu\, \mathrm{CR}_0 - \nu\mathrm{CR}(t).$$

By using the theory of monotone ordinary differential equations [26], we deduce that the map $\tau_i \to \mathrm{CR}(t_i)$ is monotone increasing, and we get the following result.

**Theorem 5.2.** *Let assumptions 1.1, 4.1 and 5.1 be satisfied. Let $I_0$ be fixed. Then we can find a unique sequence $\tau_0, \tau_1, \ldots, \tau_n$ of non-negative numbers such that $t \to \mathrm{CR}(t)$ the solution of (3.2) fits exactly the data at any time $t_i$, that is to say that*

$$\mathrm{CR}(t_i) = \mathrm{CR}_{\mathrm{Data}}(t_i), \forall i = 1, \ldots, n+1,$$

*if and only if the following two conditions are satisfied for each $i = 0, 1, \ldots, n+1$,*

$$\mathrm{CR}_{\mathrm{Data}}(t_i) \geq \mathrm{e}^{-\nu(t_i - t_{i_1})}\mathrm{CR}_{\mathrm{Data}}(t_{i-1}) + \int_{t_{i-1}}^{t_i} \nu\, \mathrm{e}^{-\nu(t_i - \sigma)}\, \mathrm{d}\sigma(f(I_0 + S_0[1 - \Pi_{i-1}^{\mathrm{Data}}]) + \mathrm{CR}_0), \tag{5.4}$$

*where*

$$\Pi_{i-1}^{\mathrm{Data}} = \exp\left(-\sum_{j=0}^{i-2} \frac{\tau_j}{\nu f}[\mathrm{CR}_{\mathrm{Data}}(t_{j+1}) - \mathrm{CR}_{\mathrm{Data}}(t_j)]\right) \tag{5.5}$$

and

$$\mathrm{CR}_{\mathrm{Data}}(t_i) \leq \mathrm{e}^{-\nu(t_i - t_{i_1})} \mathrm{CR}_{\mathrm{Data}}(t_{i-1}) + \int_{t_{i-1}}^{t_i} \nu \, \mathrm{e}^{-\nu(t_i - \sigma)} \, \mathrm{d}\sigma (f(I_0 + S_0) + \mathrm{CR}_0). \tag{5.6}$$

**Remark 5.3.** The above theorem means that the data are identifiable for this model SI if and only if the conditions (5.4) and (5.6) are satisfied. Moreover, in that case, we can find a unique sequence of transmission rates $\tau_i \geq 0$ which gives a perfect fit to the data.

# 6. Numerical simulations

In this section, we propose a numerical method to fit the day-by-day rate of transmission. The goal is to take advantage of the monotone property of $\mathrm{CR}(t)$ with respect to $\tau_i$ on the time interval $[t_i, t_{i+1}]$. Recently, more sophisticated methods were proposed by Bakhta *et al.* [20] by using several types of approximation methods for the rate of transmission.

We start with the simplest algorithm 1 in order to show the difficulties to identify the rate of transmission.

**Algorithm 1**
**Step 1:** *We fix $S_0 = 1.4 \times 10^9$, $\nu = 0.1$ or $\nu = 0.2$ and $f = 0.5$. We consider the system*

$$\left.\begin{array}{l} S'(t) = -\tau S(t)I(t), \\ I'(t) = \tau S(t)I(t) - \nu I(t) \\ \mathrm{CR}'(t) = \nu f I(t), \end{array}\right\} \tag{6.1}$$

*and*

*on the interval of time $t \in [t_0, t_1]$. This system is supplemented by initial value $S(t_0) = S_0$ and $I(t_0) = I_0$ is given by formula (2.4) (if we consider the data only at the early stage) or formula (4.3) (if we consider all the data) and $\mathrm{CR}(t_0) = \mathrm{CR}_{\mathrm{Data}}(t_0)$ is obtained from the data.*

*The map $\tau \to \mathrm{CR}(t_1)$ being monotone increasing, we can apply a bisection method to find the unique value $\tau_0$ solving*

$$\mathrm{CR}(t_1) = \mathrm{CR}_{\mathrm{Data}}(t_1).$$

*Then we proceed by induction.*
**Step i:** *For each integer $i = 1, \ldots, n$ we consider the system*

$$\left.\begin{array}{l} S'(t) = -\tau S(t)I(t), \\ I'(t) = \tau S(t)I(t) - \nu I(t) \\ \mathrm{CR}'(t) = \nu f I(t), \end{array}\right\} \tag{6.2}$$

*and*

*on the interval of time $t \in [t_i, t_{i+1}]$. This system is supplemented by initial values $S(t_i)$ and $I(t_i)$ obtained from the previous iteration and with $\mathrm{CR}(t_i) = \mathrm{CR}_{\mathrm{Data}}(t_i)$ obtained from the data.*

*The map $\tau \to \mathrm{CR}(t_i)$ being monotone increasing, we can apply a bisection method to find the unique value $\tau_i$ solving*

$$\mathrm{CR}(t_i) = \mathrm{CR}_{\mathrm{Data}}(t_i).$$

In figure 6, we plot an example of such a perfect fit, which is the same for $\nu = 0.1$ and $\nu = 0.2$. In figure 7, we plot the rate of transmission obtained numerically for $\nu = 0.2$ in (*a*) and $\nu = 0.1$ in (*b*). This is an example of a negative rate of transmission. Figure 7 should be compared to figure 4 which gives a similar result.

In figures 8–10, we use algorithm 1 and we plot the rate of transmission obtained by using the reported cases of COVID-19 in China where the parameters are fixed as $f = 0.5$ and $\nu = 0.2$. In figures 8–10, we observe an oscillating rate of transmission which is alternately positive and negative back and forth. These oscillations are due to the amplification of the error in the numerical method itself. In figure 8, we run the same simulation as in figure 9 but during a shorter period. In figure 8, we can see that the slope of $\mathrm{CR}(t)$ at the $t = t_i$ between 2 days (the black dots) is amplified 1 day to the next.

In figure 10, we first smooth the original cumulative data by using the Matlab function $\mathrm{CR}_{\mathrm{Data}} = $ smoothdata($\mathrm{CR}_{\mathrm{Data}}$,'gaussian',50) to regularize the data and we apply algorithm 1. Unfortunately, smoothing the data does not help to solve the instability problem in figure 10.

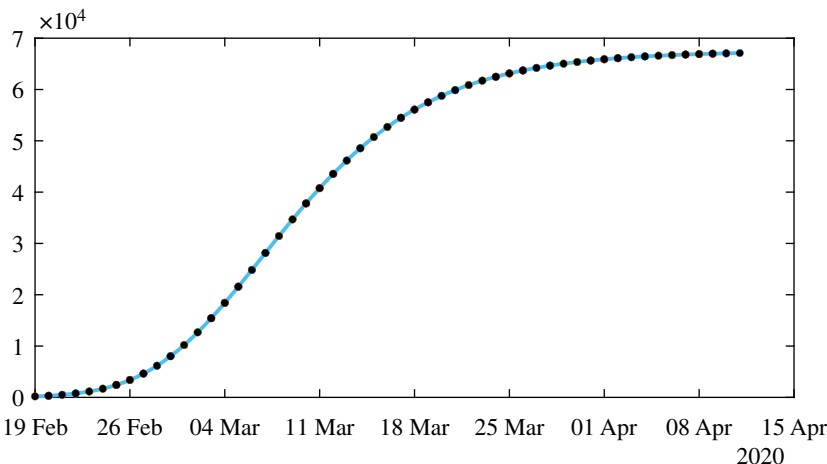

**Figure 6.** In this figure, we plot the perfect fit to the cumulative number of reported cases of COVID-19 in China. We fix the parameters $f = 0.5$ and $v = 0.2$ or $v = 0.1$ and we apply our algorithm 1 to obtain the perfect fit. The black dots correspond to data for the cumulative number of reported cases and the blue curve corresponds to the model.

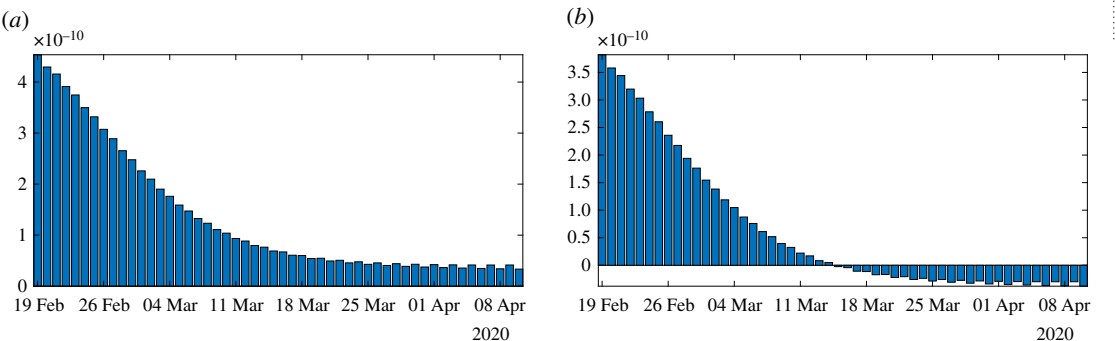

**Figure 7.** In this figure, we plot the rate of transmission obtained for the reported cases of COVID-19 in China with the parameters $f = 0.5$ and $v = 0.2$ in ($a$) and $v = 0.1$ in ($b$). This rate of transmission corresponds to the perfect fit obtained in figure 6.

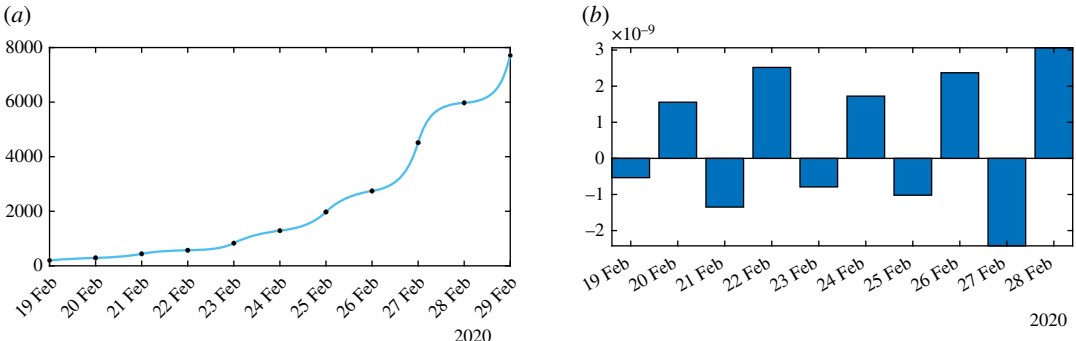

**Figure 8.** In ($a$), we plot the cumulative number of reported cases obtained from the data (black dots) and the model (blue curve). In ($b$), we plot the daily rate of transmission obtained by using algorithm 1. We see that we can fit the data perfectly. But the method is very unstable. We obtain a rate of transmission that oscillates from positive to negative values back and forth.

We need to introduce a correction when choosing the next initial value $I(t_i)$. In algorithm 1, the errors are due to the following relationship:

$$CR'(t) = vfI(t),$$

which is not respected at the points $t = t_i$ which should be reflected by the algorithm.

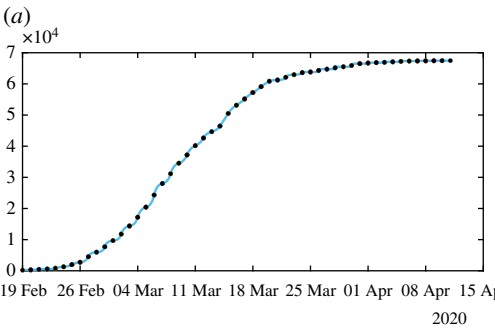
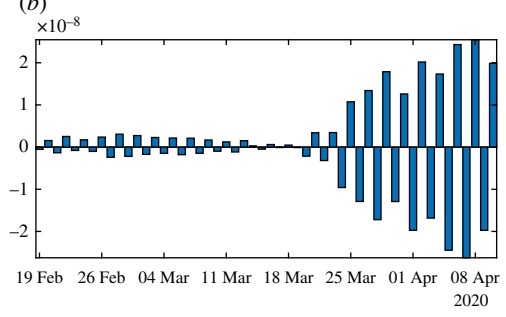

**Figure 9.** In (*a*), we plot the cumulative number of reported cases obtained from the data (black dots) and the model (blue curve) on a period six times longer than in figure 8. In (*b*), we plot the daily rate of transmission obtained by using algorithm 1. We see that we can fit the data perfectly. But the method is very unstable like on figure 8. We obtain a rate of transmission that oscillates from positive to negative values back and forth.

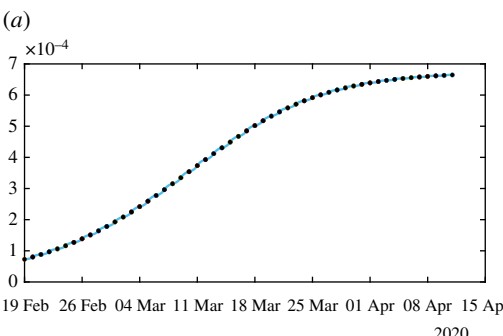
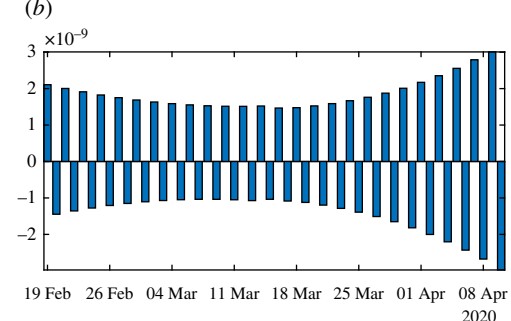

**Figure 10.** We apply algorithm 1 to the regularized data. In (*a*), we plot the regularized cumulative number of reported cases obtained from the data (black dots) and the model (blue curve). In (*b*), we plot the daily rate of transmission obtained by using algorithm 1. We see that we can fit the data perfectly. But the method is very unstable. We obtain a rate of transmission that oscillates from positive to negative values back and forth.

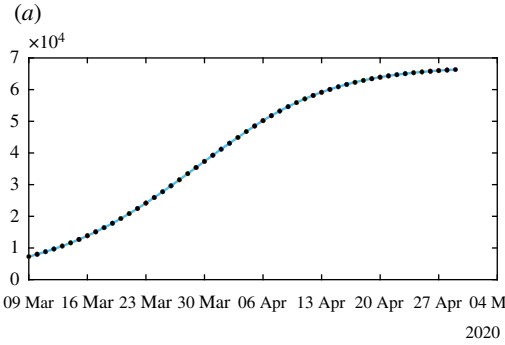
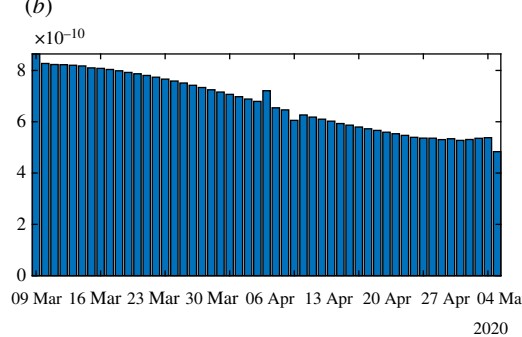

**Figure 11.** In this figure, we plot the rate of transmission obtained by using the reported cases of COVID-19 in China with the parameters $f = 0.5$ and $\nu = 0.2$. We first regularize the data by applying the Matlab function $CR_{Data} = \text{smoothdata}(CR_{Data},$ 'gaussian',50). Then we apply algorithm 2 to the regularized data. In (*a*), we plot the regularized cumulative number of reported cases obtained after smoothing (black dots) and the model (blue curve). In (*b*), we plot the daily rate of transmission obtained by using algorithm 2. We see that we can fit the data perfectly and this time the rate of transmission is becoming reasonable.

In figure 11, we smooth the data first by using the Matlab function $CR_{Data} = \text{smoothdata}(CR_{Data},$ 'gaussian',50), and we apply algorithm 2 by approximating equation (6.6) by

$$I_i = \frac{[CR_{Data}(t_i) - CR_{Data}(t_{i-1})]}{(\nu \times f)}. \tag{6.3}$$

In figure 11, we no longer observe the oscillations of the rate of transmission.

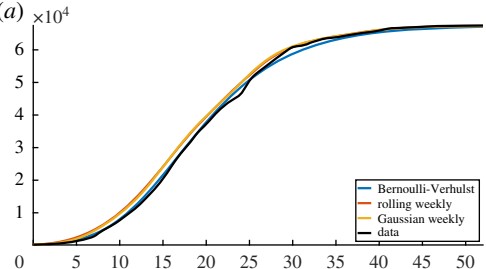
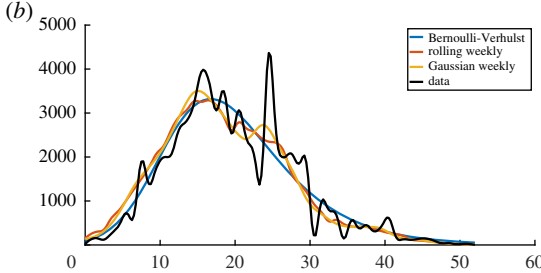

**Figure 12.** In this figure, we plot the cumulative number of reported cases (*a*) and the daily number of reported cases (*b*). The black curves are obtained by applying the cubic spline Matlab function 'spline(Days,DATA)' to the cumulative data. The left-hand side is obtained by using the cubic spline function and right-hand side is obtained by using the derivative of the cubic spline interpolation. The blue curves are obtained by using cubic spline function to the day-by-day values of cumulative number of cases obtained from the best fit of the Bernoulli–Verhulst model. The orange curves are obtained by computing the rolling weekly daily number of cases (we use the Matlab function 'smoothdata(DAILY,'movmean',7)') and then by applying the cubic spline function to the corresponding cumulative number of cases. The yellow curves are obtained by using the Gaussian weekly smoothing to the daily number of cases (we use the Matlab function 'smoothdata(DAILY,'gaussian',7)') and then by applying the cubic spline function to the corresponding cumulative number of cases.

**Algorithm 2**

*We fix $S_0 = 1.4 \times 10^9$, $\nu = 0.1$ or $\nu = 0.2$ and $f = 0.5$. Then we fit the data by using the method described in §2 to estimate the parameters $\chi_1$, $\chi_2$ and $\chi_3$ from day 1 to 10. Then we use*

$$
\left.
\begin{aligned}
S_0 &= 1.40005 \times 10^9, \\
I_0 &= \chi_2\, \chi_1 \frac{[\exp(\chi_2\,(t_0 - 1))]}{(f\,\nu)} \\
CR_0 &= \chi_1\, \exp(\chi_2\, t_0) - \chi_3.
\end{aligned}
\right\}
\tag{6.4}
$$

*and*

*For each integer $i = 0, \ldots, n$, we consider the system*

$$
\left.
\begin{aligned}
S'(t) &= -\tau S(t)I(t), \\
I'(t) &= \tau S(t)I(t) - \nu I(t) \\
CR'(t) &= \nu f I(t),
\end{aligned}
\right\}
\tag{6.5}
$$

*and*

*for $t \in [t_i, t_{i+1}]$. Then the map $\tau \to CR(t_{i+1})$ being monotone increasing, we can apply a bisection method to find the unique $\tau_i$ solving*

$$
CR(t_{i+1}) = CR_{Data}(t_{i+1}).
$$

*The key idea of this new algorithm is the following correction on the I-component of the system. We start a new step by using the value $S(t_i)$ obtained from the previous iteration and*

$$
I_i = CR'_{Data} \frac{(t_i)}{(\nu f)}
\tag{6.6}
$$

*and*

$$
CR_i = CR_{Data}(t_i).
\tag{6.7}
$$

In figure 12, we plot several types of regularized cumulative data in (*a*) and several types of regularized daily data in (*b*). Among the different regularization methods, an important one is the Bernoulli–Verhulst best-fit approximation.

In figure 13, we plot the rate of transmission $t \to \tau(t)$ obtained by using algorithm 2. We can see that the original data give a negative transmission rate while at the other extreme the Bernoulli–Verhulst seems to give the most regularized transmission rate. In figure 13*a*, we observe that we now recover almost perfectly the theoretical transmission rate obtained in §4. In figure 13*b*, the rolling weekly average regularization and in figure 13*c* the Gaussian weekly average regularization still vary a lot and in both cases, the transmission rate becomes negative after some time. In figure 13*c*, the original data give a transmission rate that is negative from the beginning. We conclude that it is crucial to find a 'good' regularization of the daily number of cases. So far the best regularization method is obtained by using the best fit of the Bernoulli–Verhulst model.

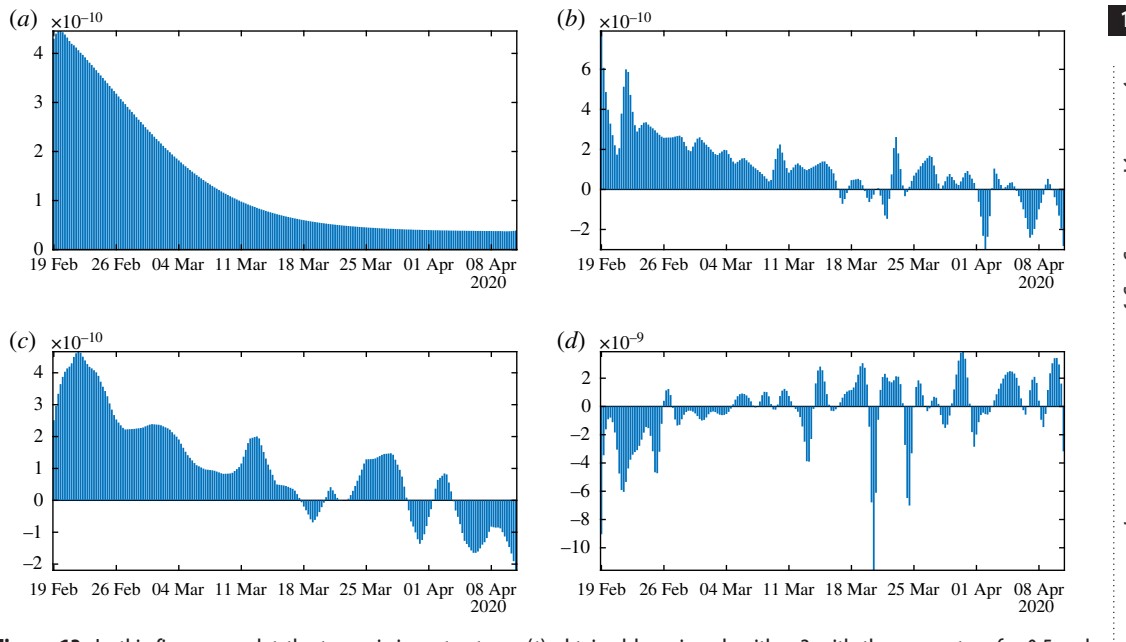

**Figure 13.** In this figure, we plot the transmission rates $t \to \tau(t)$ obtained by using algorithm 2 with the parameters $f = 0.5$ and $v = 0.2$. We use the cumulative data obtained by using (a) the Bernoulli–Verhulst regularization, (b) the rolling weekly average regularization, (c) the Gaussian weekly average regularization and in (d) we use the original cumulative data.

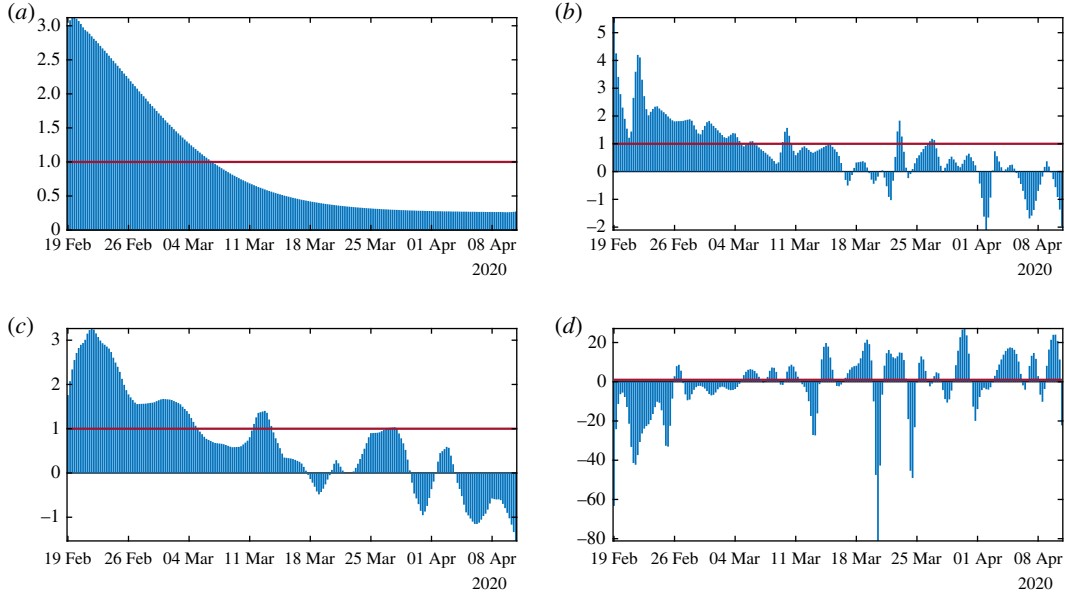

**Figure 14.** In this figure, we plot the daily basic reproduction number $t \to R_0(t) = \tau(t)S(t)/v$ obtained by using algorithm 2 with the parameters $f = 0.5$ and $v = 0.2$. We use the cumulative data obtained by using (a) the Bernoulli–Verhulst regularization, (b) the rolling weekly average regularization, (c) the Gaussian weekly average regularization and in (d) we use the original cumulative data.

**Remark 6.1.** For each simulation figure 13b,c, it is possible to obtain a transmission rate $t \to \tau(t)$ that is non-negative for all time $t$ by increasing sufficiently the parameter $v$. Nevertheless, we do not present these simulations here because the corresponding values of $v$ to obtain a non-negative $\tau(t)$ are unrealistic.

In figure 14(a–d respectively), we plot the daily basic reproduction number corresponding to the figure 13(a–d respectively). The red line corresponds to $R_0 = 1$. We see some complex behaviour for figure 14b,c,d is again unrealistic.

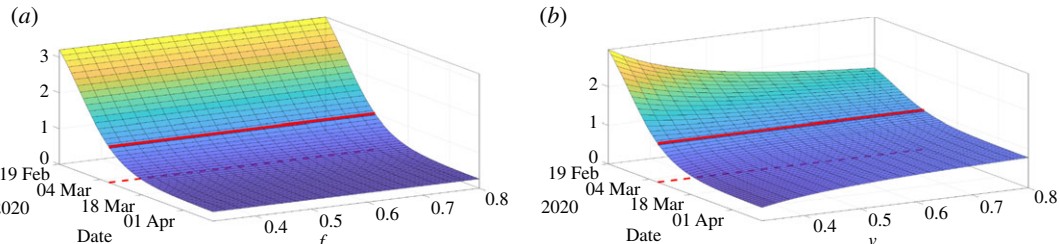

**Figure 15.** In this figure we plot $R_0(t) = \tau(t)S(t)/v$ the daily basic reproduction number and we vary the parameter $f$ (a) and $v$ (b).

## 7. Discussion

Estimating the parameters of an epidemiological model is always difficult and generally requires strong assumptions about their value and their consistency and constancy over time. Despite this, it is often shown that many sets of parameter values are compatible with a good fit of the observed data. The new approach developed in this article consists first of all in postulating a phenomenological model of growth of infectious, based on the very classic model of Verhulst, proposed in demography in 1838 [28]. Then, obtaining explicit formulae for important parameter values such as the transmission rate or the initial number of infected (or for lower and/or upper limits of these values), gives an estimate allowing an almost perfect reconstruction of the observed dynamics.

The uses of phenomenological models can also be regarded as a way of smoothing the data. Indeed, the errors concerning the observations of new infected cases are numerous:

— the census is rarely regular and many countries report late cases that occurred during the weekend and at varying times over-add data from specific counts, such as those from homes for the elderly;
— the number of cases observed is still underestimated and the calculation of not-reported new cases of infected is always a difficult problem [21];
— the raw data are sometimes reduced for medical reasons of poor diagnosis or lack of detection tools, or for reasons of domestic policy of states.

For all these causes of error, it is important to choose the appropriate smoothing method (moving average, spline, Gaussian kernel, auto-regression, generalized linear model, etc.). In this article, several methods were used and the one which allowed the model to perfectly match the smoothed data was retained.

In this article, we developed several methods to understand how to reconstruct the rate of transmission from the data. In §2, we reconsidered the method presented in [21] based on an exponential fit to the early data. The approach gives a first estimation of $I_0$ and $\tau_0$. In §3, we prove a result to connect the time-dependent cumulative reported data and the transmission rate. In §4, we compare the data to the Bernoulli–Verhulst model and we use this model as a phenomenological model. The Bernoulli–Verhulst model fits the data for mainland China very well. Next by replacing the data by the solution of the Bernoulli–Verhulst model, we obtain an explicit formula for the transmission rate. So we derive some conditions on the parameters for the applicability of the SI model to the data for mainland China. In §5, we discretized the rate of transmission and we observed that given some daily cumulative data, we can get at most one perfect fit the data. Therefore, in §6, we provide two algorithms to compute numerically the daily rates of transmission. Such numerical questions turn out to be a delicate problem. This problem was previously considered by another French group, Bakhta *et al.* [20]. Here we use some simple ideas to approach the derivative of the cumulative reported cases combined with some smoothing method applied to the data.

To conclude this article, we plot the daily basic reproduction number

$$R_0(t) = \frac{\tau(t)S(t)}{v}$$

as a function of the time $t$ and the parameters $f$ or $v$. The above simple formula for $R_0$ is not the real basic reproductive number in the sense of the number of newly infected produced by a single infectious. But this is a simple formula which gives a tendency about the growth or decay of the number of infectious. In figure 15a, the daily basic reproduction number is almost independent of $f$, while in figure 15b, $R_0(t)$ is depending on $v$ mostly for the small value of $v$. The red curve on each surface in figure 15 corresponds to

the turning point (i.e. time $t \geq t_0$ for which $R_0(t) = 1$). We also see that turning point is not depending much on these parameters.

Concerning contagious diseases, public health physicians are constantly facing four challenges. The first concerns the estimation of the average transmission rate. Until now, no explicit formula had been obtained in the case of the SIR model, according to the observed data of the epidemic, that is to say the number of reported cases of infected patients. Here, from realistic simplifying assumptions, a formula is provided (formula (4.5)), making it possible to accurately reconstruct theoretically the curve of the observed cumulative cases. The second challenge concerns the estimation of the mean duration of the infectious period for infected patients. As for the transmission rate, the same realistic assumptions make it possible to obtain an upper limit to this duration (inequality (4.8)), which makes it possible to better guide the individual quarantine measures decided by the authorities in charge of public health. This upper bound also makes it possible to obtain a lower bound for the percentage of unreported infected patients (inequality (4.8)), which gives an idea of the quality of the census of cases of infected patients, which is the third challenge faced by epidemiologists, specialists of contagious diseases. The fourth challenge is the estimation of the average transmission rate for each day of the infectious period (dependent on the distribution of the transmission over the 'ages' of infectivity), which will be the subject of further work and which poses formidable problems, in particular those related to the age (biological age or civil age) class of the patients concerned. Another interesting prospect is the extension of methods developed in the present paper to the contagious non-infectious diseases (i.e. without causal infectious agent), such as social contagious diseases, the best example being that of the pandemic linked to obesity [29–31], for which many concepts and modelling methods remain available.

Data accessibility. The data in my paper are public and can be found at: https://en.wikipedia.org/wiki/COVID-19_pandemic_in_mainland_China; http://www.nhc.gov.cn/yjb/pzhgli/new_list.shtml.

Authors' contributions. P.M. conceived and designed the study, and analysed the data. P.M. and Q.G. carried out the analysis and performed numerical simulations, and all authors conducted the literature review. All authors participated in writing and reviewing of the manuscript.

Competing interests. The authors declare no conflict of interest.

Funding. This research was funded by the Agence Nationale de la Recherche in France (Project name: MPCUII (P.M.) and (Q.G.)).

# Appendix A. Supplementary table

We use cumulative reported data from the National Health Commission of the People's Republic of China and the Chinese CDC for mainland China. Before 11 February, the data were based on confirmed testing. From 11 February to 15 February, the data included cases that were not tested for the virus, but were clinically diagnosed based on medical imaging showing signs of pneumonia. There were 17 409 such cases from 10 February to 15 February. The data from 10 February to 15 February specified both types of reported cases. From 16 February, the data did not separate the two types of reporting, but reported the sum of both types. We subtracted 17 409 cases from the cumulative reported cases after 15 February to obtain the cumulative reported cases based only on confirmed testing after 15 February. The data are given in table 1 with this adjustment.

**Table 1.** Cumulative data describing confirmed cases in mainland China from 20 January to 18 March 2020. The data are taken from [22–24].

**January**

| 19 | 20 | 21 | 22 | 23 | 24 | 25 |
|----|----|----|----|----|----|----|
| 198 | 291 | 440 | 571 | 830 | 1287 | 1975 |
| 26 | 27 | 28 | 29 | 30 | 31 | |
| 2744 | 4515 | 5974 | 7711 | 9692 | 11 791 | |

**February**

| 1 | 2 | 3 | 4 | 5 | 6 | 7 |
|----|----|----|----|----|----|----|
| 14 380 | 17 205 | 20 438 | 24 324 | 28 018 | 31 161 | 34 546 |
| 8 | 9 | 10 | 11 | 12 | 13 | 14 |
| 37 198 | 40 171 | 42 638 | 44 653 | 46 472 | 48 467 | 49 970 |
| 15 | 16 | 17 | 18 | 19 | 20 | 21 |
| 51 091 | 70 548–17 409 | 72 436–17 409 | 74 185–17 409 | 75 002–17 409 | 75 891–17 409 | 76 288–17 409 |
| 22 | 23 | 24 | 25 | 26 | 27 | 28 |
| 76 936–17 409 | 77 150–17 409 | 77 658–17 409 | 78 064–17 409 | 78 497–17 409 | 78 824–17 409 | 79 251–17 409 |
| 29 | | | | | | |
| 79 824–17 409 | | | | | | |

**March**

| 1 | 2 | 3 | 4 | 5 | 6 | 7 |
|----|----|----|----|----|----|----|
| 79 824–17 409 | 79 824–17 409 | 79 824–17 409 | 80 409–17 409 | 80 552–17 409 | 80 651–17 409 | 80 695–17 409 |
| 8 | 9 | 10 | 11 | 12 | 13 | 14 |
| 80 735–17 409 | 80 754–17 409 | 80 778–17 409 | 80 793–17 409 | 80 813–17 409 | 80 824–17 409 | 80 844–17 409 |
| 15 | 16 | 17 | 18 | | | |
| 80 860–17 409 | 80 881–17 409 | 80 894–17 409 | 80 928–17 409 | | | |

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
