## [Reviewer comments · Royal Society Open Science]

Review History

Decision letter (RSOS-201878.R0)

Dear Dr Magal:

It is a pleasure to accept your manuscript entitled "SI epidemic model applied to COVID-19 data in mainland China" in its current form for publication in Royal Society Open Science. The comments of the reviewer(s) who reviewed your manuscript are included at the foot of this letter.

COVID-19 rapid publication process:

We are taking steps to expedite the publication of research relevant to the pandemic. If you wish, you can opt to have your paper published as soon as it is ready, rather than waiting for it to be published the scheduled Wednesday.

This means your paper will not be included in the weekly media round-up which the Society sends to journalists ahead of publication. However, it will still appear in the COVID-19 Publishing Collection which journalists will be directed to each week (<https://royalsocietypublishing.org/topic/special-collections/novel-coronavirus-outbreak>).

If you wish to have your paper considered for immediate publication, or to discuss further, please notify openscience_proofs@royalsociety.org and press@royalsociety.org when you respond to this email.

on behalf of Dr Jianhong Wu (Associate Editor) and Dr Glenn Webb (Subject Editor).

Associate Editor Dr Jianhong Wu Comments to Author:

Associate Editor
Comments to the Author:
Congratulation for a fine piece of modelling work addressing a critical issue.

Some small editorial changes may be done during the production. This includes

1. Avoiding, in the abstract, direct reference to papers such as Liu et al [8]. This can be replaced by, for example, Liu et al. *Biology*, 2020.
2. Also, some minor changes can be made during proofreading, such as "methods" rather than "method" in the first sentence of the conclusion.
3. Finally, in the introduction, I would NOT say that virulence declines due to mutation. I would simply that virulence may change over time due to mutation.
